# A Spectroscopy Solution for Contactless Conductivity Detection in Capillary Electrophoresis

**DOI:** 10.3390/mi15121430

**Published:** 2024-11-28

**Authors:** Tomas Drevinskas, Audrius Maruška, Hirotaka Ihara, Makoto Takafuji, Linas Jonušauskas, Domantas Armonavičius, Mantas Stankevičius, Kristina Bimbiraitė-Survilienė, Elzbieta Skrzydlewska, Ona Ragažinskienė, Yutaka Kuwahara, Shoji Nagaoka, Vilma Kaškonienė, Loreta Kubilienė

**Affiliations:** 1Instrumental Analysis Open Access Centre, Faculty of Natural Sciences, Vytautas Magnus University, 44404 Kaunas, Lithuania; tomas.drevinskas@gmail.com (T.D.); domantas.armonavicius@vdu.lt (D.A.); mantas.stankevicius@vdu.lt (M.S.); kristina.bimbiraite-surviliene@vdu.lt (K.B.-S.); vilma.kaskoniene@vdu.lt (V.K.); 2Faculty of Advanced Science and Technology, Kumamoto University, Kumamoto 860-8555, Japan; ihara@kumamoto-u.ac.jp (H.I.); takafuji@kumamoto-u.ac.jp (M.T.); kuwahara@kumamoto-u.ac.jp (Y.K.); 3Vital3D Technologies, Sauletekio al. 15, 10224 Vilnius, Lithuania; linas@vital3d.eu; 4Department of Analytical Chemistry, Medical University of Białystok, 15-222 Białystok, Poland; elzbieta.skrzydlewska@umb.edu.pl; 5Botanical Garden, Vytautas Magnus University, 46324 Kaunas, Lithuania; ona.ragazinskiene@vdu.lt; 6Kumamoto Industrial Research Institute, Kumamoto University, Kumamoto 860-8555, Japan; nagaoka@kumamoto-u.ac.jp; 7Department of Pharmacognosy, Medical Academy, Lithuanian University of Health Sciences, 44307 Kaunas, Lithuania; loreta.kubiliene@lsmuni.lt

**Keywords:** capacitively coupled contactless conductivity detection, electrical impedance spectroscopy, capillary electrophoresis

## Abstract

This paper introduces a novel contactless single-chip detector that utilizes impedance-to-digital conversion technology to measure impedance in the microfluidic channel or capillary format analytical device. The detector is designed to operate similarly to capacitively coupled contactless conductivity detectors for capillary electrophoresis or chromatography but with the added capability of performing frequency sweeps up to 200 kHz. At each recorded data point, impedance and phase-shift data can be extracted, which can be used to generate impedance versus frequency plots, or phase-shift versus frequency plots. Real and imaginary parts can also be calculated from the data, allowing for the generation of Nyquist diagrams. This detector represents the first of its kind in the contactless conductivity class to provide spectrum-type data, as demonstrated in capillary electrophoresis experiments.

## 1. Introduction

Capacitively coupled contactless conductivity detection (C^4^D) is a widely used and powerful technique in analytical chemistry, known for its design simplicity, low cost, and applicability to capillary and microchip formats. Many research groups rely on C^4^D for its ease of use and versatility in detecting different classes of analytes [1,2].

The detector operates by applying a signal with a selected frequency between a few kHz and a few MHz and a voltage up to a few hundred volts to the working electrode, which can be a cylinder for capillary or a pad for microchip format [3,4]. The working electrode is separated from a sensing electrode of the same size. The gap that separates both electrodes is called the detection gap. The signal is capacitively coupled from the working to the sensing electrode via the solution contained in the capillary just under the electrodes, allowing for the passage of an fA-uA level current through the detection gap. The current is then amplified, rectified, filtered, buffered, and converted from analogue to digital before being recorded and stored by appropriate electronics [5,6].

Dielectric spectroscopy and electrical impedance spectroscopy are other analytical techniques that utilize frequency-based amplitude and phase-shift measurements.

Several versions of the C^4^D detectors have been reported in the literature, each with slight variations from the basic mechanism described above. High-voltage (HV) C^4^D employs excitation voltages in the range of several hundred volts, and the sensing electronics are adapted accordingly [7]. Resonance-based C^4^D utilizes a capacitor–resistor network to capture signals within a resonance zone, enhancing the detector’s sensitivity [8]. Meanwhile, single-chip detectors utilize capacitance-to-digital conversion technology to record the charge transferred from the working electrode to the sensing electrode [9,10]. Also, a space flight, radiation-hardened version of a multi-head C^4^D has been reported in the literature [11].

The C^4^D detectors can be categorized as single-frequency or variable-frequency and single-voltage-level or variable-voltage-level instruments, depending on the excitation source. The excitation signal shape is typically either sine-wave or square-wave.

The limitations of C^4^D detectors arise from the dependence on excitation voltage and frequency. Increasing the voltage can enhance sensitivity but may also cause signal saturation and limit detector readings. Similarly, higher frequency can increase current transfer and sensitivity but is also limited by saturation and frequency bandwidth. Thus, optimizing voltage and frequency for specific analytical needs is crucial, and multi-functional excitation sources may offer better detection characteristics. Alternatively, single-frequency and single-voltage-level C^4^Ds are smaller and more suitable for integration in portable and autonomous analytical systems [12,13].

C^4^Ds are commonly used in capillary electrophoresis (CE) and ion chromatography, allowing the detection and determination of charged or chargeable substances [14,15]. Other applications include bubble detection and, more recently, measurement of inorganic solvents and their mixtures [10]. However, dielectric spectroscopy, a similar technique in principle, has been used for decades in the measurement of inorganic mixtures [16].

A significant limitation of C^4^D has been the lack of qualitative information on the analyzed sample compared to techniques such as UV-Vis spectroscopy, fluorimetry, or mass spectrometry. For this reason, we undertook this research with the primary aim of developing a method to add a spectral component to C^4^D detection.

The detector described in this study has the advantage of recording the absolute values of the impedance, in contrast to C^4^D systems that only provide ratio-metric values based on the excitation signal’s amplitude. Furthermore, C^4^D systems require additional signal conditioning steps, such as full wave rectifying and baseline offsetting, which can introduce errors in the measurements [17]. In C^4^D measurements, conductivity is calculated based on the amplitude of the signal, while capacitance-to-digital conversion technology calculates capacitance based on the accumulated and measured charge in the reference capacitor. In some instances, the C^4^D signal is expressed in arbitrary units (a. u.) or calibrated as conductivity, while capacitance-to-digital conversion expresses units in farads. However, these calculations must be performed using complex mathematics as described in electrical impedance spectroscopy theory, and the proportional calculations used in C^4^D are only semi-accurate. This work involves hardware development, and we aimed to design a capacitively coupled, contactless conductivity-based detector suitable for capillary format separations, especially capillary electrophoresis.

## 2. Materials and Methods

### 2.1. Chemicals

Sodium hydroxide (99.0%) was purchased from Reachem (Petržalka, Slovakia); ascorbic acid and L-Histidine (His) (99.0%) were purchased from Alfa Aesar (Karlsruhe, Germany); L-Ascorbic acid (99.7%), vanillic acid (97.0%), ferulic acid (99.0%), and syringic acid (95.0%) were purchased from SigmaAldrich (St. Louis, MO, USA); methanol (p.a.) was purchased from Chempur (Piekary Śląskie, Poland); gallic acid (95.0%) was purchased from Thermo Scientific (Waltham, MA, USA); and rutin (95.0%) was purchased from Merck (Darmstadt, Germany). Bidistilled water was prepared in the laboratory with a Fistreem Cyclon bidistillator (Cambridge, UK).

### 2.2. Detector Design

We acquired the integrated circuit (IC) AD5933 from Analog Devices (Wilmington, MA, USA), while other electrical components, connectors, and wires were procured locally. The Arduino UNO development board was obtained from Arduino (Torino, Italy).

We designed the detector’s printed circuit board (PCB) with the KiCAD V4.0.5. electrical engineering software (KiCAD team) and fabricated it in-house using a custom-built milling machine. The electrodes were fashioned from medical-grade stainless steel syringe needles, cut to a length of 20 mm, and soldered onto the PCB pads. To make soldering easier, we coated the outer surface of the stainless-steel needle with solder. For this, we utilized a specialized soldering flux.

### 2.3. Procedures

The HP3DCE system (Agilent Technologies, Waldbronn, Germany) was employed for CE separations. The detector’s PCB was incorporated into an HP3DCE cassette, and the separation capillary was installed, as demonstrated in previous papers [9]. The total length of the capillary (L_tot_) was 58 cm, while the effective length (Leff) to the detector was 46 cm. Separation was carried out using a 50 mM ascorbic acid solution. Samples were injected at 50 mbar for 10 s, and separations were performed at 25 °C at 15 kV. Prior to injection, the capillary was washed with 100 mM NaOH solution for 5 min, water for 3 min, and then with a 50 mM ascorbic acid solution for 5 min.

### 2.4. Detector Operation

The communication between the detector and Arduino UNO was established using the inter-integrated circuit (I2C) protocol (Figure 1), and data was transmitted from the Arduino to the computer via a USB cable and serial port. The detector was powered by a 5 V line from the Arduino UNO microcontroller. The power supply was downregulated with TC1014 IC (Microchip, Tampa, FL, USA) to 3.3 V and used to supply AD5933 IC.

The actuator electrode pin was connected to the detector, and the AD5933 supplied the excitation signal of predetermined amplitude and frequency. The signal passed through the capacitively coupled electrodes and the background electrolyte inside the separation capillary before reaching the sensing part of the AD5933. A feedback resistor of 560 kΩ was included for first-stage signal amplification.

The microcontroller was programmed using the Arduino prototyping platform to facilitate communication with the AD5933 IC. The algorithm provided in the AD5933 datasheet was used to create the sketch [18]. This involved implementing the necessary steps for the device to perform the measurement, such as setting the start frequency, the number of increments, and the frequency increment width and sending these parameters into specific AD5933 registers.

Subsequently, the device was initialized, and the frequency sweep started after putting it into standby mode. Once the signal measurement was completed and converted into a digital format, the data registers within AD5933 IC were read. The frequency value was then incremented, and the entire sweep, read, and increment process was repeated until the entire frequency range was measured. Finally, the device was powered down.

Throughout the analysis time, the device repeated the listed cycles continuously during the electrophoretic separation process, and one spectrum was recorded every 1.05 s.

To regulate the operation of the AD5933, a simplified graphical user interface (GUI) was programmed as it has been for the past work [9]. The GUI could send instructions to the microcontroller, allowing the system to operate automatically. It consisted of several panels representing real-time data, including Bode and Nyquist plots that displayed the data of the measured impedance at specific times, as well as a main plot that monitored the impedance of the selected frequency over time.

### 2.5. Polyphenol Detection Procedure

A miniaturised CE-C^4^D system (Figure 2), described in our previous work [13], was used to obtain calibration curves of vanillic, syringic, ferulic, gallic acids, and rutin. Calibration solutions were prepared at concentrations of 0.1, 0.25, 0.5, 0.75, and 1.0 µg/mL, dissolved in 100% methanol.

Analysis was performed using fused silica capillaries with a total length of 25cm and an effective length of 10 cm. Separation was carried out using a 100% methanol solution. Samples were injected at 50 mbar for 5 s, and separations were performed at room temperature (25 °C) at 15 kV. Prior to the injection, the capillary was washed with 100% methanol solution for 5 min.

## 3. Results and Discussion

### 3.1. Response of Different Compounds

The experimental setup of this section is described in Figure 2A. Different solutions were flowed through the separation capillary. It is known that C^4^D responds well to the compounds that have charge or are chargeable. Also, as clarified in our past work with capacitance-to-digital C^4^D systems and known in dielectric spectroscopy, different solvents and their mixtures show different responses [10,16]. Therefore, for this work, we were interested in how phenolic acids can be detected with our technique.

When measured 1 µg/mL solutions of vanillic, syringic, ferulic, gallic acids, and rutin indicated capacitance (fF) values of 29.7, 10.0, 22.0, 1.5, and 6.4, correspondingly, and when translated into impedance (MΩ), the values were 401, 1275, 318, 7105, and 2925. To translate into the impedance, we used the equation Z = 1/(2πfC), where Z is impedance (Ω), π is 3.14, f is frequency (Hz, 32,000), and C is capacitance (F).

Also, for further investigations in our hardware design work, we selected a model that would exhibit lower impedances. For that, we selected NaOH, L-ascorbic acid, and bidistilled water. We prepared 100 mM of NaOH solution in water and 50 mM of L-ascorbic acid solution in water and measured. While rinsing the separation capillary with each solution, we measured these impedances (at 32,000 Hz): 7.8 MΩ, 20.7 MΩ, and 15.3 MΩ for the 100 mM NaOH, bidistilled water, and the 50 mM L-ascorbic acid solution. For further optimizations, we used the latter solutions.

### 3.2. Design and Optimization

Compared to capacitance-to-digital conversion systems, impedance-to-digital conversion systems require feedback resistors to determine the amplification of the measured signal. In typical C^4^Ds, a feedback resistor is used in trans-impedance amplifiers as well. In this method, a known amplitude signal is fed into the unknown impedance, and the resulting signal is amplified in an integrated trans-impedance amplifier. The feedback resistor value sets the gain of the integrated inverting operation amplifier circuit. If the value of the feedback resistor is known, the impedance value can be easily calculated by measuring the amplified signal’s amplitude and using a simple proportion.

In addition to the resistive component, phase shift, which reflects the capacitive component, is also crucial. Without knowledge of phase shift, it is impossible to determine whether the measured signal passed through a resistive, capacitive, or inductive load. To address this issue, Bode plots are helpful in observing phase shift, while Nyquist plots can be used to determine the equivalent model (a combination of resistors and capacitors) of the electrochemical system being measured. Hence, we can conclude that the Nyquist plot is a valuable tool for providing qualitative information about the electrochemical system. To obtain information related to phase shift, the AD5933 IC is equipped with an integrated Digital Fourier Transform core.

Initial attempts to record a signal produced successful measurements. However, although the measured signal reflected a reasonable overall impedance, its phase shift was inaccurate, and obtaining sensitive Nyquist plots was impossible. As a result, the device required calibration with a reference that had impedance and phase-shift values not within the measurement range, but that was not too different from the electrochemical system.

To improve the accuracy of the device, a new calibration procedure was developed using known electrical components. The following steps were taken: (1) The electrodes were disconnected from the detector, (2) a 10 MΩ resistor and a 1.2 pF capacitor were soldered in parallel to the impedance measurement position, (3) the CE cassette containing the detector was inserted into the CE system, (4) the temperature was set to 25 °C, and (5) a file containing known values of calculated impedance, phase shift, and real and imaginary parts was uploaded to the AD5933 IC calibration register. The calculation of the impedance and phase shift for the 10 MΩ resistor and 1.2 pF capacitor is straightforward and well-defined in electrical impedance spectroscopy theory. During the calibration mode, the device adjusted the gain factor and stored reference phase-shift values in its registers based on the calculated values and the measured real signal and phase shift.

With proper calibration, impedance-to-digital conversion technology can provide precise measurements of absolute impedance by considering both the electrodes and the liquid under measurement. Moreover, this technique has the potential to reveal the exact equivalent circuit of the C^4^D electrode configuration.

### 3.3. Measurements of the Solutions

The experimental setup of this section is described in Figure 2B. The capillary was flushed with multiple solutions while the detector recorded the signal. The first solution used was 100 mM NaOH, followed by distilled water and 50 mM ascorbic acid. Figure 3 shows the change in baseline at different measurement frequencies as the solutions were introduced into the capillary.

The detector generates spectral data that includes impedance and phase-shift information at different excitation frequencies and time points, allowing for the extraction of time-series data and the display of specific parameters (Figure 3A–C).

Analysis of the detector’s response reveals that the lowest impedance was observed with the 100 mM NaOH solution, while the highest impedance was recorded for the bidistilled water. Interestingly, the impedance values obtained for the 50 mM vitamin C solution did not follow the same trend as those observed for the 100 mM NaOH solution and bidistilled water.

The Nyquist plots obtained for each solution revealed more detailed information than a comparison of baseline recordings at different frequencies, as shown in Figure 2, D. Nyquist plots provide a detailed spectrum for each solution, revealing distinct differences. Interestingly, the 50 mM ascorbic acid solution showed a nearly linear response across different frequencies between the real and imaginary parts of impedance, while distilled water and 100 mM NaOH solutions had a breakpoint in their spectra.

Additionally, the real-part impedance was smaller than the imaginary-part impedance for all solutions, indicating that current flow was not primarily limited by the resistive path in the electrode geometry. In this case, the resistive path represents the conductance of the solution inside the capillary between the electrodes. High values of the negative imaginary part suggest that capacitive coupling between the contactless electrode and solution in the capillary is a limiting factor in current transfer. It means that the current passing through the electrode–solution–electrode path was primarily limited by capacitance.

Once more, the breakpoints observed in the Nyquist plots of distilled water and 100 mM sodium hydroxide solution resemble the beginning of a semi-oval geometry, which suggests a parallel combination of a resistor and capacitor. This may indicate that the current in the solution is not solely transferred through the conductance of the electrolyte but through a different path. However, to confirm this, a more comprehensive study with a wider range of investigated frequencies is required, which can only be achieved with improved circuitry in the current detector.

### 3.4. CE Separation and Detection of Cations

The experimental setup of this section is described in Figure 2C. We successfully recorded Na^+^ and histidine peaks during CE separation (Figure 4) and extracted frequency spectrum information (Figure 4D) of those peaks using impedance-to-digital conversion technology. However, as expected, the IDC showed lower sensitivity compared to a typical C^4^D detector due to lower resolution acquiring data, as IDC has a 12-bit ADC while C^4^D typically uses a 24-bit acquisition. Furthermore, we employed the IDC in the simplest possible configuration without a buffering stage on the excitation signal and sensing part. Despite this, peak shape and signal-to-noise ratio were similar between the different frequency electropherograms, which is expected and has been reported in multiple works where different excitation frequencies are only used as a tool for optimizing detection parameters.

Nyquist plots were extracted for both Na^+^ and histidine peaks (Figure 4D). Although there is a slight difference in the plots between the two peaks, the plot presents values greater than 10 MOhm for the real part and a range greater than 30 MΩ for the imaginary part, causing the spectra to be zoomed out and difficult to visualize using classical techniques. The fact that the impedance in the imaginary and real parts covers a big range (from several kΩ to multiple MΩ), it may be worth investigating augmented visualization techniques so that visual identification of analytes can be achieved.

To facilitate comparing all acquired data, plots of impedance versus frequency and phase shift versus frequency were generated (Figure 5).

Different solutions (100 mM NaOH, distilled water, and 50 mM L-ascorbic acid) are clearly distinguishable in both plots, with the phase-shift versus frequency plot showing more pronounced differences. However, in Figure 5B, a phase shift lower than −90° is observed, indicating an error likely due to the measurement of very high impedances at that frequency. Additionally, for comparison, peaks from CE separation were included in the plot. The difference between the 1.6 mM Na^+^ peak and the 0.4 mM L-histidine peak in 50 mM L-ascorbic acid BGE is negligible.

Our proposed technique enables the scanning and recording of spectra over time. To analyse the data, we averaged 50 baseline spectra (50 mM L-ascorbic acid) from the electropherogram and 10 spectra corresponding to the tip of each peak (1.6 mM Na^+^ and 0.4 mM L-histidine). Since the differences across the range investigated are more significant than the minor variations between individual spectra, we used the averaged baseline data as a reference and subtracted it from each peak. Reference subtraction was performed for both the impedance and phase shift of each peak. The results are shown in Figure 6.

In both plots (impedance and phase shift) shown in Figure 6, the 1.6 mM Na^+^ and 0.4 mM L-histidine peaks are clearly distinguishable across the entire measured frequency range (5–195 kHz). The impedance profile aligns with C^4^D theory. However, the phase-shift plot (Figure 6B) reveals several discrepancies where the patterns of the two peaks do not match. For instance, at 35 kHz, the Na^+^ peak scatter line shows a local maximum, whereas the L-His peak scatter line does not follow the same trend. Similarly, at 65 kHz, the Na^+^ peak scatter line indicates another local maximum, while the L-His peak scatter line shows a local minimum. At 85 kHz, both peaks exhibit local minima, and at 155 kHz, both peaks display local maxima. The change shift at this level of detail for separated peaks in CE has not been previously investigated. With the hardware described in this work now available for reproduction, these findings highlight the need for further exploration. However, the authors acknowledge that such observations could potentially be attributed to noise.

Additionally, our work could greatly benefit miniaturised gas chromatography instrumentation [19,20,21]. A miniaturised gas chromatographic instrument has been developed by Qin et al. and successfully demonstrated to operate using the same capacitance-to-digital-based C^4^D technology employed in our initial miniaturised CE studies [9,10].

## 4. Conclusions

A novel detector that provides spectral data has been developed, and its proof-of-concept has been demonstrated successfully. The detector can function as a standalone instrument or be integrated into a capillary electrophoresis system. It can record impedance and phase shift and provide time-series data, which is essential for generating Bode or Nyquist plots. Nyquist plots can be derived from any data point, including peaks that represent different analytes in electropherograms.

## Figures and Tables

**Figure 1 micromachines-15-01430-f001:**
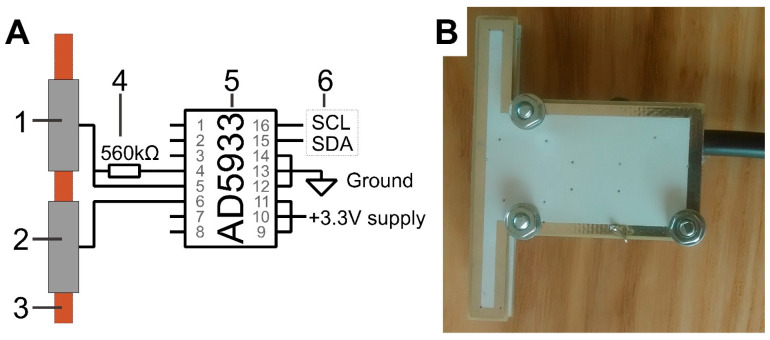
Designed detector. (**A**) Simplified schematic diagram of the detector. (**B**) Photograph of the detector. Annotations: 1—sensing electrode, 2—excitation electrode, 3—separation capillary, 4—feedback resistor, 5—AD5933 IC, 6—I2C communication connections.

**Figure 2 micromachines-15-01430-f002:**
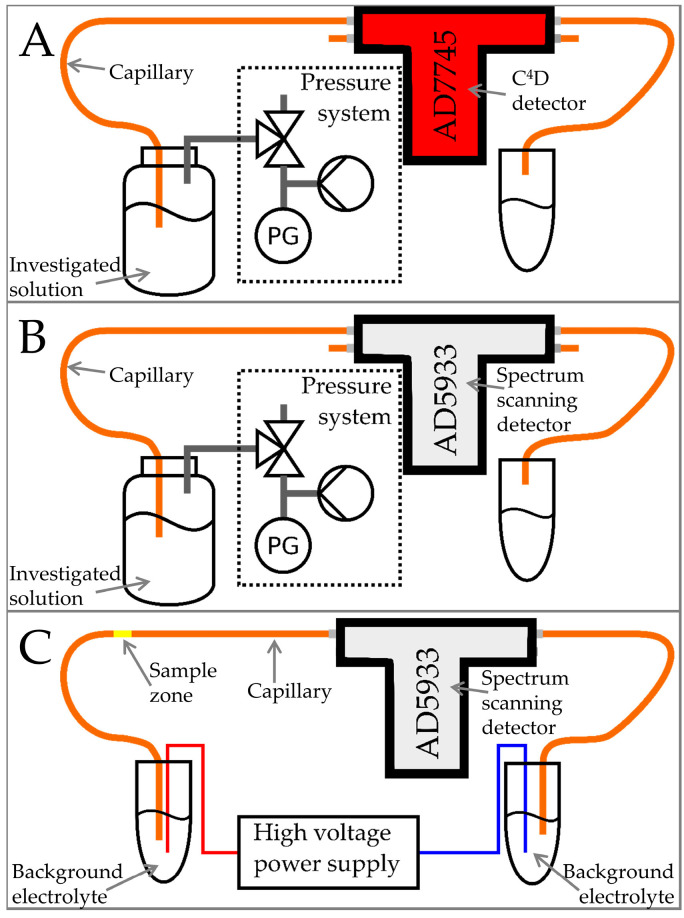
Experiment design schematic diagram. Solution measurements with capacitance-to-digital based C^4^D detector (**A**). Solution measurements with the spectrum scanning detector proposed in this work (**B**). Capillary electrophoresis experiment with the spectrum scanning detector proposed in this work (**C**).

**Figure 3 micromachines-15-01430-f003:**
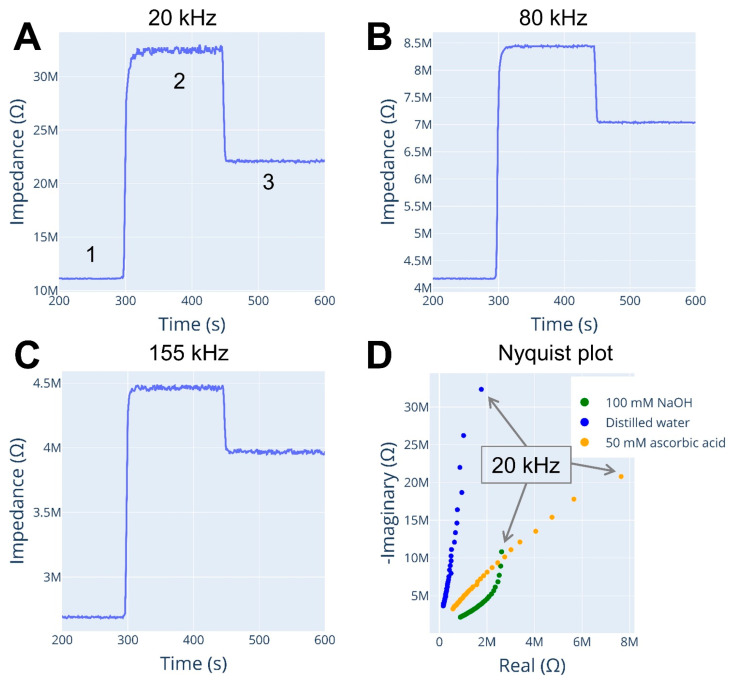
Detector response during capillary rinsing procedure. Time series at excitation of (**A**) 20 kHz, (**B**) 80 kHz, (**C**) 155 kHz. (**D**) Nyquist plots represent a spectrum of each rinsed solution. Annotations: 1—100 mM NaOH, 2—distilled water, 3—50 mM ascorbic acid. Fused silica capillary used: 50 µm I.D., 365 µm O.D., 58 cm L_tot_, 46 cm L_eff_. Sweep: 20–190 kHz frequency, 3.3 V_pp_ amplitude.

**Figure 4 micromachines-15-01430-f004:**
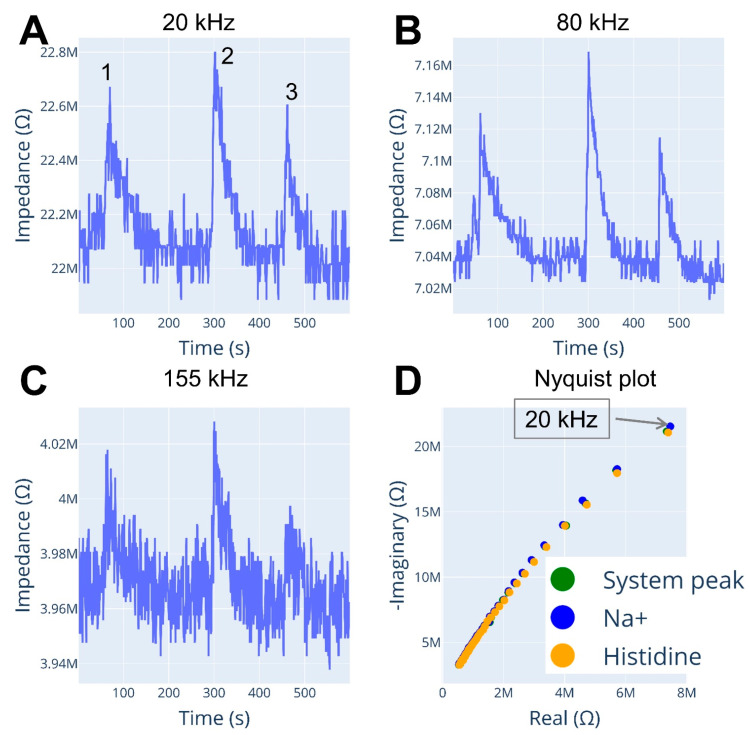
Electropherograms and peak spectra. Recorded at an excitation frequency of (**A**) 20 kHz, (**B**) 80 kHz, (**C**)155 kHz. (**D**) Nyquist plot of each peak in the electropherogram. Annotations: 1—system peak, 2—1.6 mM Na^+^, 3—0.4 mM L-Histidine. Voltage potential: 15kV. Capillary: 58 cm L_tot_, 46 cm L_eff_. Background electrolyte—50 mM L-ascorbic acid. Sweep: 20–180 kHz frequency, 3.3 V_pp_ amplitude.

**Figure 5 micromachines-15-01430-f005:**
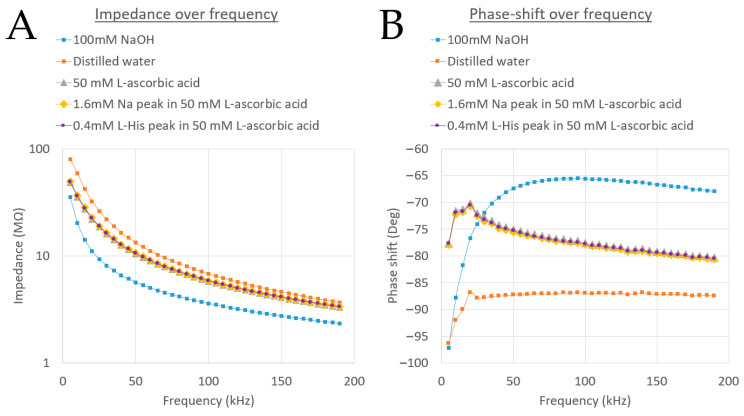
Spectral data for different solutions: impedance versus frequency (**A**) and phase-shift versus frequency (**B**).

**Figure 6 micromachines-15-01430-f006:**
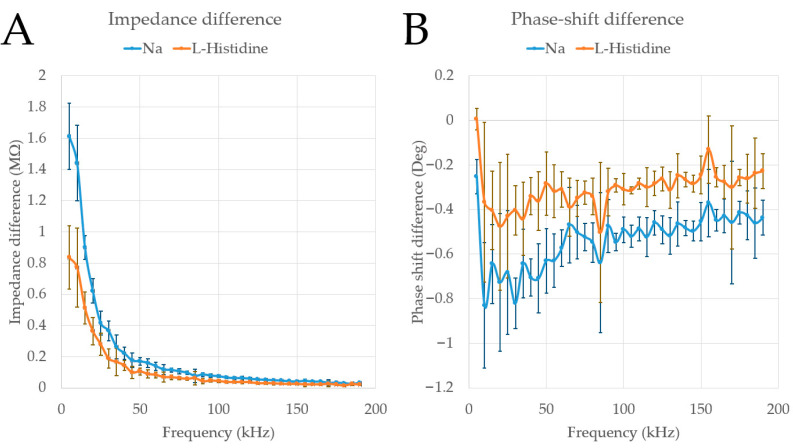
Spectral data for different peaks referenced to the baseline: (**A**) Impedance difference versus frequency and (**B**) phase-shift difference versus frequency.

## Data Availability

The original contributions presented in the study are included in the article, further inquiries can be directed to the corresponding author.

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
