# Peer review of "A Spectroscopy Solution for Contactless Conductivity Detection in Capillary Electrophoresis"

_micromachines, 2024, doi:10.3390/mi15121430_

Round 1

Reviewer 1 Report

Comments and Suggestions for Authors

In this work the authors introduced a novel contactless single-chip detector that utilizes impedance-to-digital conversion technology to measure impedance in the microfluidic channel or capillary format analytical device. The main question addressed by the paper is try to introduce a  novel contactless single-chip detector that utilizes impedance-to-digital conversion technology to measure impedance in the microfluidic channel or capillary format analytical device. The device testing was detailly described with well-established ways for testing and presenting results (Bode's and Nyquist's plots). Due the fact that searching on Web of Science gives only a handful of papers of this topic, it can be stated need for such investigations. On the other hand, it can be seen a gap in the field too.

The introduction was well written and gave good insight about scientific interest. On the other hand, it must be stated lack of similar papers and investigations. It can be found only a handful of papers by searching on Web of Science. The authors made a very detailed description of used materials and analytical methods. During the read, I felt able to repeat the experimental work without the problem. Used literature is a mixture. It is neccessary to have six self-cited papers? I'm aware of similar papers lack, but you have almost one third of self-cited papers. The conclusion is consistent to the evidence given in the manuscript.

Generally, I find presentation quality very good.  This manuscript can be considered for publication in Micromachines after minor revison (spell check and reconsdering slef-cited papers). 

Reviewer 2 Report

Comments and Suggestions for Authors

The Introduction is not clearly written. Lines 195-210 seem to explain this somewhat but this section should be moved to the Introduction. The idea to provide some qualitative ID for this type of detector is a good one and the design and construction of this detector seems inexpensive. This following sentence in quotes should not be at the end but moved much earlier in the Introduction. "Dielectric spectroscopy and electrical impedance spectroscopy are other analytical techniques that utilize frequency-based amplitude and phase shift measurements."  However, I do not understand why this work is analogous to spectroscopy as indicated in the title.  Spectroscopy can distinguish very similar molecules. The plots in Figure 2d are for very different solutions.  For example, could this method distinguish between a carboxylic acid and an alcohol? Why is there a system peak in Figure 3a? Linear regression analysis equations and correlation coefficients should be provided for Na and histidine based on the data in Figure 3c. Qualitative identification of the Na and histidine analytes using Nyquist plots is not shown.

Author Response

Please find responses attached
